# Influence of Psychological Biomarkers on Therapeutic Adherence by Patients with Non-Alcoholic Fatty Liver Disease: A Moderated Mediation Model

**DOI:** 10.3390/jcm10102208

**Published:** 2021-05-20

**Authors:** Jesús Funuyet-Salas, Agustín Martín-Rodríguez, María Ángeles Pérez-San-Gregorio, Manuel Romero-Gómez

**Affiliations:** 1Department of Personality, Assessment and Psychological Treatment, Faculty of Psychology, University of Seville, 41018 Seville, Spain; amartinr@us.es (A.M.-R.); anperez@us.es (M.Á.P.-S.-G.); 2UCM Digestive Diseases and Ciberehd, Virgen del Rocío University Hospital, Institute of Biomedicine of Seville, University of Seville, 41013 Seville, Spain; mromerogomez@us.es

**Keywords:** NAFLD, therapeutic adherence, self-efficacy, quality of life, social support, depressive symptoms, physical activity, Mediterranean diet

## Abstract

Our aim was to analyze whether depressive symptoms mediated the association between physical quality of life (QoL) and adherence to physical activity in patients with non-alcoholic fatty liver disease (NAFLD), as well as the association between social support and adherence to diet. We also examined whether self-efficacy exerted a moderating role in these associations. QoL (SF-12), social support (MSPSS), depressive symptoms (HADS), self-efficacy (GSE), physical activity (IPAQ) and diet (MEDAS) were evaluated in 413 biopsy-proven NAFLD patients. Mediation and moderated mediation models were conducted using the SPSS PROCESS v3.5 macro. Results showed that depressive symptoms mediated the relationship between physical QoL and adherence to physical activity (indirect effect = 6.248, CI = 1.917–10.727), as well as the relationship between social support and adherence to diet (indirect effect = 0.148, CI = 0.035–0.275). Self-efficacy also moderated the indirect effects of QoL and social support on therapeutic adherence through depressive symptoms. Specifically, the higher self-efficacy was, the lower the negative impact on the NAFLD patient’s mental health. In conclusion, self-efficacy is defined as a protective factor for therapeutic adherence by NAFLD patients with a psychosocial risk profile. Self-efficacy should, therefore, be a main psychological target in future multidisciplinary NAFLD approaches.

## 1. Introduction

Today’s predominant lifestyle, based on sedentarism and a diet high in saturated fats and sugars, has brought on severe metabolic consequences. Non-alcoholic fatty liver disease (NAFLD), closely related to metabolic diseases, such as type 2 diabetes mellitus (T2DM) and obesity, in recent years, has become the main cause of chronic liver disease [1,2]. It is even expected to emerge in the short-term future as the main cause of liver transplantation in the world, which makes NAFLD an alarming public health problem [3].

NAFLD patients must comply with a therapeutic plan based on changing their lifestyle. The combined plans of the European Association for the Study of the Liver, European Association for the Study of Diabetes and the European Association for the Study of Obesity (EASL-EASD-EASO) recommend 150–200 min of moderately intense aerobic physical activity for a total of three to five weekly sessions. Resistance training is also effective. In addition, they also recommend a Mediterranean diet, which restricts foods with a high saturated fat or sugar content and promotes foods rich in monounsaturated fatty acids and omega-3 [4,5]. Compliance with these plans is necessary to lose 10% of their body weight, which is fundamental to achieve a reduction in steatosis, improvement in inflammation or regression of liver fibrosis [6,7]. However, therapeutic adherence is inadequate in more than half of NAFLD patients [8]. This could be partly explained by the lack of willpower these patients commonly have for changing [9] and the effects that certain psychological biomarkers could have on following long-term therapeutic recommendations.

In this sense, NAFLD is closely associated with a negative impact on the patient’s quality of life (QoL) and mental health, mainly worse physical functioning [10,11] and more depressive symptoms than other chronic liver diseases or the general population [12,13]. The state of physical and mental health predicts regular practice of physical activity in diabetic and obese patients [14,15]; however, there is no evidence in this respect in NAFLD patients.

In addition, satisfactory social support favors successful continued behavior changes related to weight loss [16] and has been linked to a decrease in the intensity and frequency of concomitant depressive symptoms in chronic liver patients [17]. This is important, because both social support and mental health determine how a person manages stressful everyday situations, such as those derived from staying on a long-term diet [18,19]. In fact, worse dieting behavior has been found in diabetic and obese patients who report depression and insufficient social support [20,21,22,23]. The relationship between social support and mental health in NAFLD has recently been demonstrated [24]; however, its impact on therapeutic adherence in these patients is still unknown.

Finally, self-efficacy has demonstrated its importance in managing chronic illnesses, as it can confirm the patient’s perception of his/her ability to change health-related behavior [25]. To feel motivated to undertake a task, one must have confidence in the chances of success [26,27]. As less self-efficacy has been found in NAFLD patients than in other chronic liver diseases [28], this could be a critical factor in understanding the lack of motivation these patients have for their treatment. Many studies have found better performance of physical activity and diet by diabetic and obese patients who have high perceived self-efficacy [29,30,31,32]. Similar results were found by Zelber-Sagi et al. in a study with 146 ultrasound-diagnosed NAFLD patients [33]; however, more evidence is necessary to confirm the importance of self-efficacy in these patients.

In this context, we proposed testing how physical QoL, perceived social support, depressive symptoms, self-efficacy, adherence to diet and physical activity are interrelated in NAFLD patients. We also analyzed whether depressive symptoms mediate the relationship between physical QoL and adherence to physical activity, on one hand, and on the other, the relationship between perceived social support and adherence to diet. Finally, in both cases, we examined whether self-efficacy moderates these relationships.

## 2. Materials and Methods

### 2.1. Participants

We selected a group of 413 patients with biopsy-proven NAFLD (252 men and 161 women) with a mean age of 55.5 ± 11.6 years. The study was approved by the Ethics Committee of the Virgen del Rocío University Hospital of Seville. Table 1 shows the sociodemographic characteristics of the patients who gave their informed consent for their participation. This study followed the 1975 Helsinki Declaration guidelines for good clinical practice.

### 2.2. Instruments

The 12-Item Short Form Health Survey (SF-12v.2) evaluates health-related QoL in 12 items rated on three- or five-point Likert-type scales [34,35]. Using Quality Metric Health Outcomes TM Scoring Software 5.0. (QualityMetric Incorporated LLC, Johnston, RI, USA), two summary components can be calculated, the mental and the physical component summary (PCS). In this study, we only analyzed the latter because NAFLD mainly impacts on the patient’s physical quality of life [10,11]. Scores vary from 0 (worst health condition) to 100 (best heath condition), with higher scores indicating better QoL. In our sample, the Cronbach’s alpha for the different dimensions varied from 0.73 to 0.94, while for the PCS it was 0.92 [34].

The Multidimensional Scale of Perceived Social Support (MSPSS) evaluates perceived social support from family, friends and a partner or significant other in 12 items rated on a seven-point Likert-type scale [36]. The instrument provides a total scale corresponding to the average of the scores on each of its items. Scores vary from 1 to 7, with higher scores indicating more social support. We used the Spanish version of the instrument [37]. The Cronbach’s alpha was 0.95 for the total scale, while it varied from 0.96 to 0.99 for the various social support dimensions. 

The Hospital Anxiety and Depression Scale (HADS) measures anxiety and depressive symptoms in 14 items on a four-point Likert-type scale [38]. The instrument provides a total score for anxiety and another for depression. In this study, we only analyzed the latter because the NAFLD impact on the mental health of patients is mainly on depressive symptoms [12,13]. The scores vary from 0 to 21, with higher scores showing worse mental health. We used the Spanish version of the instrument [39]. The Cronbach’s alpha was 0.89 for the total depression score. 

The General Self-Efficacy Scale (GSE) evaluates the perceived ability to manage stressful situations in everyday life adequately in ten items on a ten-point Likert-type scale [40]. The instrument provides a total corresponding to the sum of scores on each of the items. Scores vary from 10 to 100 for the total score, with higher scores indicating more self-efficacy. We used the Spanish version of the instrument [41]. The Cronbach’s alpha was 0.95 for the total instrument. 

The International Physical Activity Questionnaire—Short Form (IPAQ-SF) measures the intensity, frequency and duration of an individual’s physical activity in seven items that ask about the time spent during the past week doing vigorous- and moderate-intensity activities, as well as walking and time spent sitting [42]. The instrument provides a total score corresponding to the individual’s weekly physical activity, which is calculated based on the sum of the metabolic equivalent of task (METs) per minute and week found for vigorous, moderate physical activity and for walking. High scores indicate more physical activity. Due to the scale’s characteristics, the composite reliability index was used to check its internal consistency [43], which varied from 0.91 to 0.95 for its different dimensions.

The Mediterranean Diet Adherence Screener (MEDAS) measures dietary patterns in 14 items scored from 0 to 1 by their habits and frequency of eating certain foods (olive oil, legumes, fruit and vegetables, etc.) in the traditional Mediterranean diet or not [44]. Scores vary from 0 to 14 for the total scale, with higher scores indicating better adherence to the Mediterranean diet. The Cronbach’s alpha was 0.52 for the total instrument scale. 

### 2.3. Procedure

The sample consisted of 413 NAFLD patients (Figure 1), from ten Spanish hospitals. The inclusion criteria were: over 18 years of age, diagnosed with NAFLD by liver biopsy, give their informed consent to participate, not have been diagnosed with any severe or disabling psychopathological condition and able to understand the study assessment instruments. All of the participants were assessed by the same psychologist, using the same assessment instruments and always applied in the following order: psychosocial interview, SF-12, HADS, EAG, MSPSS, IPAQ and MEDAS.

### 2.4. Statistical Analysis

The independent samples *t*-test and analysis of variance (ANOVA) were applied to examine the differences in adherence to physical activity and diet in the sample by sociodemographic (gender, age, marital status, education, employment) and clinic (NASH, significant fibrosis, type 2 diabetes, body mass index (BMI) and obesity) characteristics.

Pearson correlations were used to analyze the associations between physical QoL, perceived social support, self-efficacy, depressive symptoms, adherence to physical activity and diet.

The mediation and moderated mediation models were computed using the SPSS PROCESS macro (Version 3.5, Columbus, OH, USA) developed by Hayes [45]. First, depressive symptoms were established as mediator in the relationship between physical QoL and adherence to physical activity, using Model 4 [46]. The same model was applied to check whether the depressive symptoms mediated the relationship between perceived social support and adherence to diet. In both cases, 5000 bootstrap samples were used to test the indirect effects estimated. The effect of mediation was significant if the 95% confidence interval (CI) of the indirect effects did not contain 0. Then, Model 7 [46] was used, again with 5000 bootstrap samples, to analyze the moderated mediation effect, that is, whether self-efficacy moderated the indirect effects of physical QoL on adherence to physical activity through depressive symptoms. The same model was applied to check whether self-efficacy also moderated the indirect effects of perceived social support on adherence to diet through depressive symptoms.

The pick-a-point approach and Johnson–Neyman technique, using the PROCESS macro, checked the significance of moderation. The pick-a-point approach found three groups for the moderator, which could be classified as those participants with low, medium and high scores on this variable. In continuation, this technique calculated the conditional effect of the predictor variable on the criterion variable for each of those values, generating a confidence interval [47]. The Johnson–Neyman technique determined the regions of values in the range of the moderator variable where the effect of the predictor variable on the continuous variable was significant [47]. A two-sided *p*-value < 0.05 was considered statistically significant.

## 3. Results

### 3.1. Sociodemographic and Clinic Variables

Of the 413 participants with a mean age of 55.5 (*SD* = 11.6) and a BMI of 30.8 (*SD* = 5.2), 61% were men and 39% women, 77.7% had a partner, 44.1% had a low education, 47.9% were actively employed, 56.9% had NASH, 37.8% had significant fibrosis, 32.4% were diabetics, and 52.1% were obese. The mean score on physical activity was 925.9 (*SD* = 1130.2), while on adherence to diet, it was 8.1 (*SD* = 2.3). The results of the independent samples *t*-test and ANOVA showed significant gender (*p* = 0.016) and obesity (*p* = 0.001) differences in the scores on adherence to physical activity. The scores on adherence to diet also showed significant gender (*p* = 0.004), obesity (*p* = 0.040) and employment (*p* = 0.025) differences. Finally, there was a significant positive correlation between age and scores on adherence to diet (*p* < 0.001) and a significant negative correlation between BMI and scores on adherence to physical activity (*p* < 0.001) and diet (*p* = 0.025), as shown by the Pearson correlation analysis (Table 1).

### 3.2. Correlation Analysis

The Pearson correlation analysis (Table 2) revealed that adherence to physical activity was positively associated with physical QoL (*r* = 0.19, *p* < 0.001), while adherence to diet was positively associated with perceived social support (*r* = 0.22, *p* < 0.001). Both were also positively associated with self-efficacy (physical activity, *r* = 0.18, *p* < 0.001; diet, *r* = 0.18, *p* < 0.001) and negatively associated with depressive symptoms (physical activity, *r* = −0.19, *p* < 0.001; diet, *r* = −0.20, *p* < 0.001). Furthermore, both physical QoL and social support were positively associated with self-efficacy (QoL, *r* = 0.46, *p* < 0.001; social support, *r* = 0.56, *p* < 0.001) and negatively associated with depressive symptoms (QoL, *r* = −0.52, *p* < 0.001; social support, *r* = −0.56, *p* < 0.001). Self-efficacy was negatively associated with depressive symptoms (*r* = −0.70, *p* < 0.001).

### 3.3. Mediation Analysis

Figure 2 shows the significant indirect effect of depressive symptoms mediating in the association between physical QoL and adherence to physical activity (6.127, *p* = 0.047). The bootstrapped 95% CI did not include 0 (1.633 to 10.566), confirming the significant indirect effect of physical QoL on adherence to physical activity through depressive symptoms. The direct effect of physical QoL on the adherence to physical activity was not significant after mediation analysis (8.437, *p* = 0.190), showing full mediation of depressive symptoms.

Figure 3 shows the significant indirect effect of depressive symptoms mediating in the association between perceived social support and adherence to diet (0.136, *p* = 0.020). The bootstrapped 95% CI did not include 0 (0.022 to 0.261), confirming the significant indirect effect of social support on adherence to diet through depressive symptoms. The direct effect of perceived social support on adherence to diet was significant after mediation analysis (0.318, *p* = 0.004), showing partial mediation of depressive symptoms.

### 3.4. Moderated Mediation Analysis

The moderated mediation analyses determined whether self-efficacy moderated the effects of physical QoL and perceived social support on adherence to physical activity and diet, respectively, through depressive symptoms.

Figure 4 shows the results of the moderated mediation model for adherence to physical activity. The results revealed that self-efficacy moderated the relationship between physical QoL and depressive symptoms (*β* = 0.004, *p* < 0.001). The Johnson–Neyman technique showed that the effects of perceived social support on depressive symptoms were significant at the different levels of self-efficacy from 10 to 71.474 (56.6% of the participants), where the conditional effect of physical QoL was stronger, the lower self-efficacy was. Similarly, the pick-a-point approach (Table 3 and Figure 5) showed that the negative effects of physical QoL on depressive symptoms lessened with rising self-efficacy, measured at three levels: low self-efficacy, equivalent to a standard deviation below the mean (effect = −0.147, *p* < 0.001); medium self-efficacy, equivalent to the mean (effect = −0.061, *p* < 0.001); high self-efficacy, equivalent to a standard deviation above the mean (effect = 0.025, *p* = 0.210), where the effect of physical QoL on depressive symptoms was no longer significant.

Table 4 shows the conditional indirect effects of physical QoL on adherence to physical activity through depressive symptoms at the three self-efficacy levels. The results revealed a significant fall in conditional indirect effect as self-efficacy increased (low self-efficacy, effect = 5.043, 95% CI = 1.391 to 8.915; medium self-efficacy, effect = 2.091, 95% CI = 0.509 to 3.963; high self-efficacy, effect = −0.861, 95% CI = −3.030 to 0.534), and the conditional indirect effect at this last level of self-efficacy was not even statistically significant any longer. In the pairwise comparisons between conditional indirect effects, the bootstrapped 95% CI did not include 0 (medium level–low level of self-efficacy, effect = −2.952, 95% CI = −5.513 to −0.760; high level–low level of self-efficacy, effect = −5.903, 95% CI = −11.026 to −1.520; high level–medium level of self-efficacy, effect = −2.952, 95% CI = −5.513 to −0.760), confirming that the mediation effect was moderated by self-efficacy.

Figure 6 shows the results of the moderated mediation model for adherence to diet. The results revealed that self-efficacy moderated the relationship between perceived social support and depressive symptoms (*β* = 0.027, *p* < 0.001). The Johnson–Neyman technique showed that the effects of perceived social support on depressive symptoms were significant at the various levels of self-efficacy from 10 to 74.338 (62.1% of the participants), where the conditional effect of perceived social support was stronger, the lower self-efficacy was. Similarly, the pick-a-point approach (Table 5 and Figure 7) showed that the negative effects of perceived social support on depressive symptoms decreased as self-efficacy increased, measured at three levels: low self-efficacy, equivalent to a standard deviation below the mean (effect = −1.060, *p* < 0.001); medium self-efficacy, equivalent to the mean (effect = −0.549, *p* < 0.001); high self-efficacy, equivalent to a standard deviation above the mean (effect = −0.037, *p* = 0.832), where the effect of perceived social support on depressive symptoms was no longer significant.

Table 6 shows the conditional indirect effects of perceived social support on adherence to diet at the different self-efficacy levels through depressive symptoms. The results revealed significantly lower conditional indirect effects as self-efficacy increased (low self-efficacy, effect = 0.090, 95% CI = 0.013 to 0.177; medium self-efficacy, effect = 0.047, 95% CI = 0.006 to 0.097; high self-efficacy, effect = 0.003, 95% CI = −0.027 to 0.036), and the conditional indirect effect was not even statistically significant any longer on this level of self-efficacy. In the pairwise comparisons between conditional indirect effects, the bootstrapped 95% CI did not include 0 (medium level–low level of self-efficacy, effect = −0.043, 95% CI = −0.088 to −0.006; high level–low level of self-efficacy, effect = −0.087, 95% CI = −0.176 to −0.013; high level–medium level of self-efficacy, effect = −0.043, 95% CI = −0.088 to −0.006), confirming that the mediation effect was moderated by self-efficacy.

## 4. Discussion

The impact of doing physical activity and following a Mediterranean diet on NAFLD has been widely studied in recent years [48,49,50,51]. At the same time, the relevance of psychological biomarkers has also been recently proven in NAFLD [24,52]. However, to date, the influence of psychological biomarkers on therapeutic adherence has not been studied in these patients. Our study, therefore, examined whether physical QoL, social support, depressive symptoms and self-efficacy were significantly associated with participants performing physical activity and staying on their diet. It also examined whether depressive symptoms mediated, on one hand, the relationship between physical QoL and the adherence to physical activity, and on the other, the relationship between perceived social support and adherence to diet. In both cases, it was further found whether self-efficacy moderated these relationships.

This study found therapeutic adherence of participants to be similar to what was found in other studies on NAFLD [53,54]. The results of the univariate analyses showed significant differences in therapeutic adherence of participants by gender, employment, age, BMI and obesity. To begin with, women reported better maintenance of the Mediterranean diet and less physical activity than men, in agreement with previous studies on NAFLD [55,56]. Concerning employment, those participants who were actively employed followed the Mediterranean diet less, as also found previously [57]. In line with Giraldi et al. (2020), age significantly correlated with adherence to diet, such that older participants referred more to eating the Mediterranean diet [58]. Finally, a higher BMI and the presence of obesity were associated with less physical activity and poorer adherence to the Mediterranean diet, confirming the close link between metabolic pathology and an inadequate lifestyle [58,59].

As expected, the results showed that physical QoL, perceived social support, depressive symptoms and self-efficacy were significantly associated with therapeutic adherence by NAFLD patients. Depressive symptoms fully mediated the relationship between physical QoL and adherence to physical activity. First, poor QoL predicted stronger depressive symptoms, confirming the relationship previously found between QoL and depression in NAFLD [24,60]. Greater depression, in turn, predicted less physical activity by participants, as had previously been found in healthy individuals [61], diabetics [29] and obese [15]. Metabolic alterations characteristic of NAFLD would, thus, lead to deterioration in physical functioning, which through its impact on mental health, would interfere negatively with the patient’s participation in physical activities. This, in turn, would lead to an increase in body weight and worse state of physical and mental health in a vicious circle with negative physical and psychological consequences for the patient [62,63].

At the same time, depressive symptoms partially mediated the relationship between perceived social support and adherence to diet. Low social support directly predicted worse maintenance of the Mediterranean diet by participants, as had been proven previously in patients with T2DM [23], obesity [20] and cardiovascular disease [64]. Perception of social support could buffer or strengthen the negative effects of certain factors, such as lack of time to cook or the temptation to eat unhealthy foods, on long-term maintenance of a Mediterranean diet [64,65]. In fact, eating has been described as a “social behavior”, determined by social norms and pressures and by the influence of the closest setting [66]. In our study, low social support further predicted stronger depressive symptoms, which was also related to participants adhering less to their diet. The relationship between social support and mental health of NAFLD patients is, thus, confirmed [24], as is that depressive symptoms predict worse eating habits in adults who need to modify their lifestyle to lose weight [67].

In addition, the moderated mediation analyses revealed that self-efficacy moderated the relationship between physical QoL and adherence to physical activity, on one hand, and on the other, the relationship between perceived social support and adherence to diet, both through depressive symptoms. The indirect effects of QoL and social support on therapeutic adherence through depressive symptoms were reinforced in patients with low self-efficacy and attenuated in those with high perceived self-efficacy. According to Bandura, self-efficacy is a prerequisite for modifying behavior, since the person must be confident that he/she can carry out the action and that the results will be favorable or beneficial [26,27]. Along this line, in the Health Belief Model [68], self-efficacy is essential for patients to feel motivated to modify their long-term physical activity and eating habits. This would explain the protective role of self-efficacy in our study with regard to therapeutic adherence: the higher self-efficacy was, the lower the negative effect that low physical QoL and low social support had on doing physical activity and following the Mediterranean diet, respectively, through depressive symptoms. Thus, high self-efficacy would protect from physical discomfort and pressures or lack of support in one’s close social setting [60,69], which in turn would be associated with lower concomitant depressive symptoms [29,70,71]. Strengthening self-efficacy in NAFLD patients would then lead them to tolerate and cope more adaptively with this type of obstacle or problem and, therefore, show more optimism, commitment and engagement toward therapeutic recommendations for physical activity and eating.

The results of this study demonstrate the need to progress from the traditional lifestyle intervention model, in which the NAFLD patient is only encouraged to carry out certain therapeutic recommendations, with poor adherence results, to a collaborative intervention model [8,72]. It is important to have multidisciplinary teams comprising physicians, nurses, psychologists and nutritionists who help the patient overcome the barriers that impede behavioral change, such as low self-efficacy, poor physical or mental health or lack of social support, as corroborated in our study [60,73]. First, modification of lifestyle must be undertaken with individualized therapeutic plans designed together with the patient, in which quantifiable and specific, realistic and yet challenging goals are set, for example, walk for 20 min a day, eat only when sitting at table or lose half a kilo a week [74]. The achievement of these goals would normally be associated with a gradual increase in the patient’s perceived self-efficacy [75,76], which would strengthen therapeutic adherence. For those patients who refer to limited physical or mental health, recommended therapeutic strategies include reducing the intensity of physical activity at first and increasing it gradually as agreed with the patient; learning to manage those negative emotions linked to doing physical activity; or providing the patient with resources, such as relaxation or meditation that enable him/her to manage those stressful situations that lead him/her to eating [65,77]. Finally, intervention should include support from a significant other, such as spouse or cohabiting family member who can encourage the patient to make healthy changes in the home menu or go out for a walk, which could reinforce behavioral change and, thereby, drive a healthier lifestyle and better therapeutic adherence [78].

Our study showed some limitations. First, the low Cronbach’s alpha of the MEDAS questionnaire showed low internal consistency, as had been found in previous studies in other countries that had also used this questionnaire [79,80]. Second, the cross-sectional design of the study impeded determining the long-term evolution of the results. Third, it would be necessary to validate generalization of the results to NAFLD patients in other populations, since it would have to be analyzed whether cultural diversity in lifestyle influences the relationship found between psychological biomarkers and therapeutic adherence. Nevertheless, the large size of the study sample, comprising biopsy-proven NAFLD patients from real clinical practice in several Spanish hospitals may be considered the main strength of this study.

## 5. Conclusions

Our study confirmed for the first time the importance of psychological biomarkers such as physical QoL, social support, depressive symptoms and self-efficacy in therapeutic adherence by NAFLD patients. Depressive symptoms mediated, on one hand, the relationship between physical QoL and adherence to physical activity, and on the other, the relationship between social support and adherence to diet by participants. Furthermore, self-efficacy exerted a moderating role in the indirect effects of physical QoL and social support through depressive symptoms on adherence to physical activity and dietary recommendations. The higher patient self-efficacy was, the more these indirect effects were attenuated. Therefore, future multidisciplinary NAFLD treatments should consider the impact of these psychological biomarkers on therapeutic adherence and, especially, design therapeutic strategies to improve the self-efficacy of these patients.

## Figures and Tables

**Figure 1 jcm-10-02208-f001:**
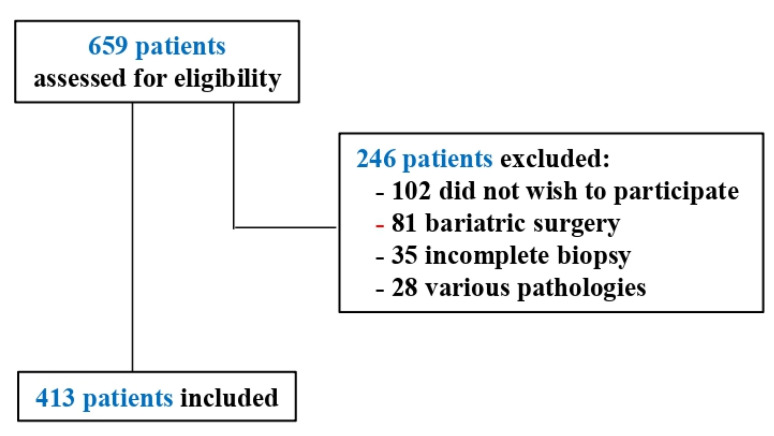
Participant selection for the study.

**Figure 2 jcm-10-02208-f002:**
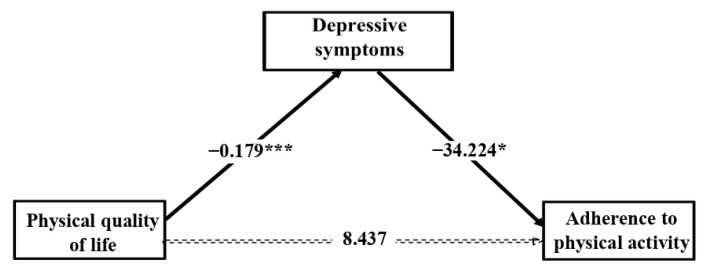
Depressive symptoms mediate the relationship between physical quality of life and adherence to physical activity. Note. The coefficients represent the indirect and direct effects estimated. Gender, BMI and obesity were entered in the analysis as covariates. * *p* < 0.05; ** *p* < 0.01; *** *p* < 0.001.

**Figure 3 jcm-10-02208-f003:**
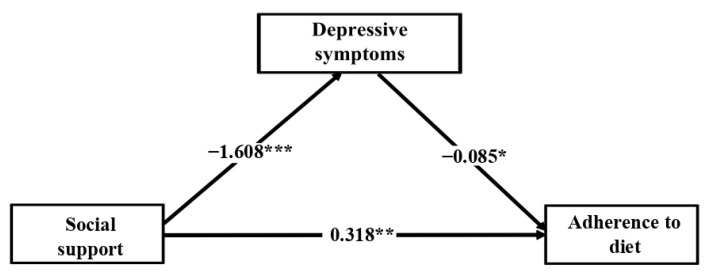
Depressive symptoms mediate the relationship between perceived social support and adherence to diet. Note. The coefficients represent the indirect and direct effects estimated. Gender, age, employment, BMI and obesity were entered in the analysis as covariates. * *p* < 0.05; ** *p* < 0.01; *** *p* < 0.001.

**Figure 4 jcm-10-02208-f004:**
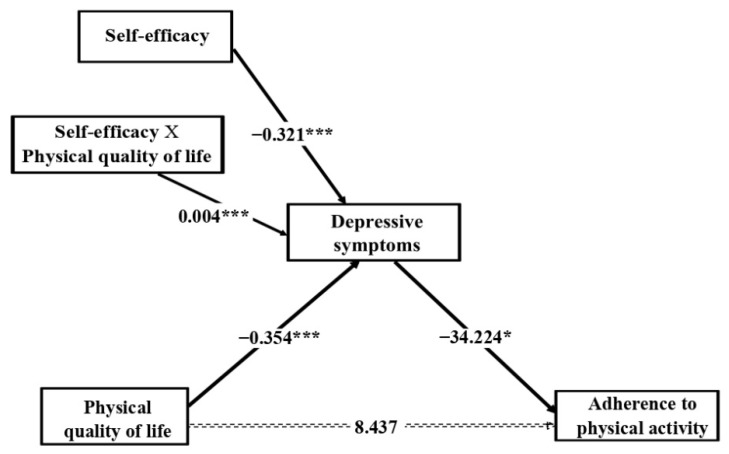
The moderating effect of self-efficacy on the relationship between physical quality of life and adherence to physical activity through depressive symptoms. Note. The coefficients represent the moderating, indirect and direct effects estimated. Gender, BMI and obesity were entered in the analysis as covariates. * *p* < 0.05; ** *p* < 0.01; *** *p* < 0.001.

**Figure 5 jcm-10-02208-f005:**
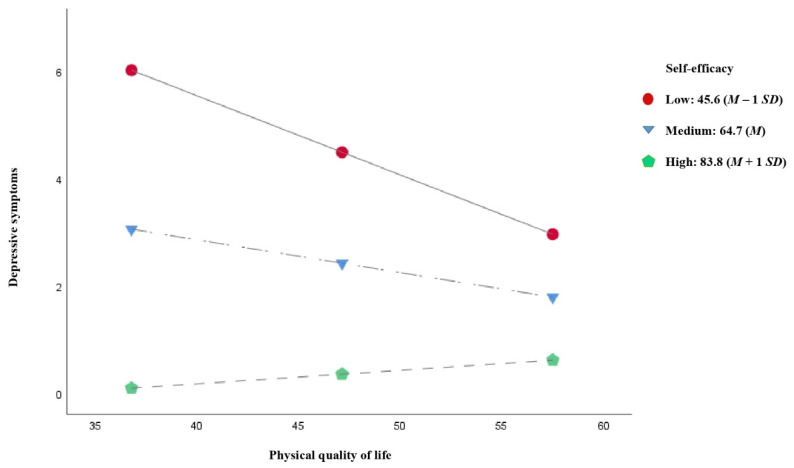
Self-efficacy moderates the effect of physical quality of life on depressive symptoms. *M*, mean; *SD*, standard deviation.

**Figure 6 jcm-10-02208-f006:**
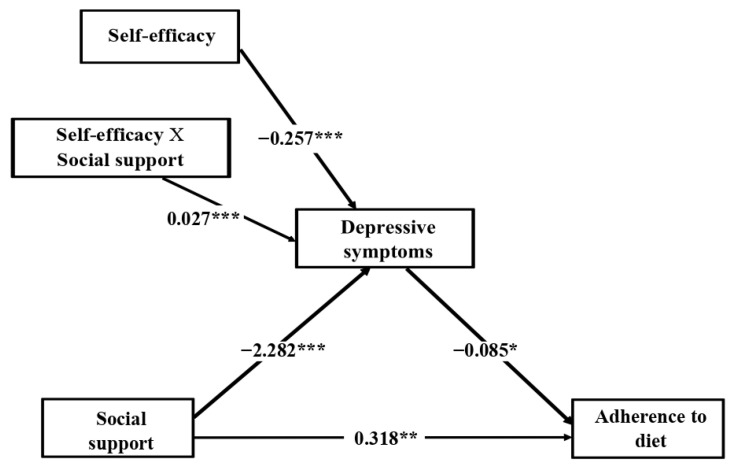
The moderating effect of self-efficacy on the relationship between perceived social support and adherence to diet through depressive symptoms. Note. The coefficients represent the moderating, indirect and direct effects estimated. Gender, age, employment, BMI and obesity were entered in the analysis as covariates. * *p* < 0.05; ** *p* < 0.01; *** *p* < 0.001.

**Figure 7 jcm-10-02208-f007:**
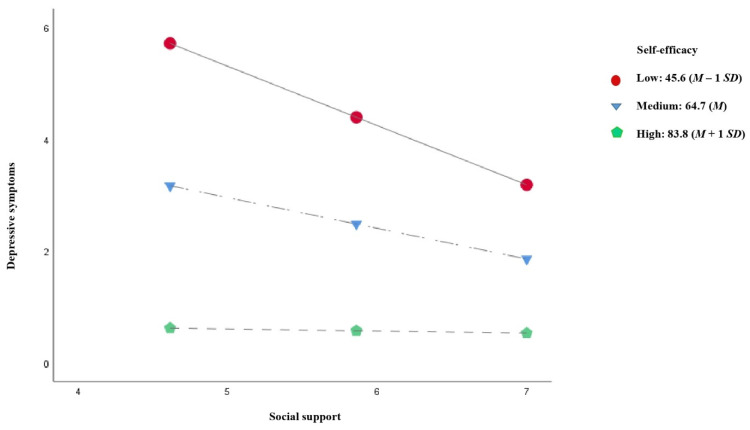
Self-efficacy moderates the effect of social support on depressive symptoms. Note. *M*, mean; *SD*, standard deviation.

**Table 1 jcm-10-02208-t001:** Univariate analyses of the differences in adherence to physical activity and diet by sociodemographic and clinic variables.

	*M (SD)*	Adherence to Physical Activity *M (SD)*	*r* (*p*)	Adherence to Diet *M (SD)*	*r* (*p*)
Age	55.1 (11.6)	925.9 (1130.2)	−0.08 (0.112)	8.1 (2.3)	0.18 (<0.001)
BMI	30.8 (5.2)	925.9 (1130.2)	−0.17 (<0.001)	8.1 (2.3)	−0.11 (0.025)
	**Total N (%)**	**Adherence to** **physical activity** ***M (SD)***	***t/F (p)***	**Adherence to** **diet** ***M (SD)***	***t/F (p)***
Gender Male Female	252 (61.0) 161 (39.0)	1027.1 (1228.5) 767.5 (938.2)	*t*(1, 398.052) = 2.43 (0.016)	7.9 (2.3) 8.5 (2.3)	*t*(1, 411) = −2.92 (0.004)
Marital status With partner Without partner	321 (77.7) 92 (22.3)	905.4 (1126.3) 997.6 (1147.3)	*t*(1, 411) = 0.69 (0.491)	8.2 (2.2) 7.9 (2.6)	*t*(1, 133.560) = −1.09 (0.279)
Education Low Medium High	182 (44.1) 118 (28.6) 113 (27.3)	883.7 (997.2) 941.8 (1241.7) 977.3 (1214.7)	*F*(2, 410) = 0.25 (0.775)	8.4 (2.2) 7.9 (2.2) 8.0 (2.6)	*F*(2, 410) = 2.50 (0.083)
Employment Working Not working	198 (47.9) 215 (52.1)	947.8 (1269.5) 905.7 (987.4)	*t*(1, 411) = 0.38 (0.706)	7.9 (2.3) 8.4 (2.3)	*t*(1, 411) = −2.25 (0.025)
NASH Absence Presence	178 (43.1) 235 (56.9)	955.4 (1210.8) 903.6 (1067.3)	*t*(1, 411) = 0.46 (0.646)	8.3 (2.4) 8.0 (2.2)	*t*(1, 411) = 1.37 (0.170)
Significant fibrosis Absence Presence	257 (62.2) 156 (37.8)	954.7 (1233.4) 878.5 (937.7)	*t*(1, 411) = 0.66 (0.508)	8.1 (2.3) 8.1 (2.3)	*t*(1, 411) = −0.09 (0.927)
Type 2 Diabetes Absence Presence	279 (67.5) 134 (32.4)	914.0 (1135.2) 950.8 (1123.6)	*t*(1, 411) = 0.31 (0.757)	8.1 (2.3) 8.2 (2.4)	*t*(1, 411) = −0.23 (0.815)
Obesity Absence Presence	198 (47.9) 215 (52.1)	1128.7 (1265.3) 739.2 (955.3)	*t*(1, 365.427) = 3.51 (0.001)	8.4 (2.3) 7.9 (2.3)	*t*(1, 411) = 2.06 (0.040)

The Pearson correlation (age, BMI), independent samples *t*-test (gender, marital status, employment, NASH, significant fibrosis, type 2 diabetes, obesity) and analysis of variance (education) were applied.

**Table 2 jcm-10-02208-t002:** Intercorrelations of adherence to physical activity, adherence to diet, physical quality of life, social support, self-efficacy and depressive symptoms.

Variables	*M* (*SD*)	1	2	3	4	5	6
1. Adherence to physical activity	925.9 (1130.2)	1	0.21 ***	0.19 ***	0.16 **	0.18 ***	−0.19 ***
2. Adherence to diet	8.1 (2.3)	0.21 ***	1	0.09	0.22 ***	0.18 ***	−0.20 ***
3. Physical quality of life	46.9 (10.5)	0.19 ***	0.09	1	0.34 ***	0.46 ***	−0.52 ***
4. Social support	5.9 (1.2)	0.16 **	0.22 ***	0.34 ***	1	0.56 ***	−0.56 ***
5. Self-efficacy	64.7 (18.8)	0.18 ***	0.18 ***	0.46 ***	0.56 ***	1	−0.70 ***
6. Depressive symptoms	2.8 (3.8)	−0.19 ***	−0.20 ***	−0.52 ***	−0.56 ***	−0.70 ***	1

* *p* < 0.05; ** *p* < 0.01; *** *p* < 0.001; Pearson correlation was applied.

**Table 3 jcm-10-02208-t003:** Effects of moderation by self-efficacy on the relationship between physical quality of life and depressive symptoms.

Self-Efficacy	Effect (*SE*)	*t (p)*	Bootstrapped 95% CI	
			Lower	Upper
Low: 45.6 (*M* − 1 *SD*)	−0.147 (0.016)	−9.06 (<0.001)	−0.179	−0.115
Medium: 64.7 (*M*)	−0.061 (0.014)	−4.34 (<0.001)	−0.089	−0.033
High: 83.8 (*M* + 1 *SD*)	0.025 (0.020)	1.25 (0.210)	−0.014	0.064

*M*: mean; *SD*: standard deviation; *SE*: standard error; CI: confidence interval; the pick-a-point approach was applied to check the significance of moderation effects.

**Table 4 jcm-10-02208-t004:** Conditional indirect effect of physical quality of life on adherence to physical activity through depressive symptoms.

	Self-Efficacy	Effect *(BootSE)*	Bootstrapped 95% CI	
			Lower	Upper
Effect 1	Low: 45.6 (*M* − 1 *SD*)	5.043 (1.957)	1.391	8.915
Effect 2	Medium: 64.7 (*M*)	2.091 (0.904)	0.509	3.963
Effect 3	High: 83.8 (*M* + 1 *SD*)	−0.861 (0.912)	−3.030	0.534
Effect 2 − Effect 1		−2.952 (1.230)	−5.513	−0.760
Effect 3 − Effect 1		−5.903 (2.460)	−10.969	−1.520
Effect 3 − Effect 2		−2.952 (1.230)	−5.513	−0.760

**Note.***M*: mean; *SD*: standard deviation; *BootSE*: bootstrap standard error; CI: confidence interval; bootstrapping was employed to analyze the conditional indirect effect.

**Table 5 jcm-10-02208-t005:** Effects of moderation by self-efficacy on the relationship between perceived social support and depressive symptoms.

Self-Efficacy	Effect (*SE*)	*t (p)*	Bootstrapped 95% CI	
			Lower	Upper
Low: 45.6 (*M* − 1 *SD*)	−1.060 (0.134)	−7.92 (<0.001)	−1.323	−0.797
Medium: 64.7 (*M*)	−0.549 (0.128)	−4.30 (<0.001)	−0.799	−0.298
High: 83.8 *(M* + 1 *SD*)	−0.037 (0.177)	−0.21 (0.832)	−0.385	0.310

**Note.***M*: mean; *SD*: standard deviation; *SE*: standard error; CI: confidence interval; the pick-a-point approach was applied to check the significance of moderation effects.

**Table 6 jcm-10-02208-t006:** Conditional indirect effect of social support on adherence to diet through depressive symptoms.

	Self-Efficacy	Effect *(BootSE)*	Bootstrapped 95% CI	
			Lower	Upper
Effect 1	Low: 45.6 (*M* − 1 *SD*)	0.090 (0.042)	0.013	0.177
Effect 2	Medium: 64.7 (*M*)	0.047 (0.023)	0.006	0.097
Effect 3	High: 83.8 (*M* + 1 *SD*)	0.003 (0.015)	−0.027	0.036
Effect 2 − Effect 1		−0.043 (0.021)	−0.088	−0.006
Effect 3 − Effect 1		−0.087 (0.042)	−0.176	−0.013
Effect 3 − Effect 2		−0.043 (0.021)	−0.088	−0.006

**Note.***M*: mean; *SD*: standard deviation; *BootSE*: bootstrap standard error; CI: confidence interval; bootstrapping was employed to analyze the conditional indirect effect.

## Data Availability

The raw data supporting the findings of this study will be made available by the corresponding author upon reasonable request.

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
