# Peer review of "Influence of Psychological Biomarkers on Therapeutic Adherence by Patients with Non-Alcoholic Fatty Liver Disease: A Moderated Mediation Model"

_jcm, 2021, doi:10.3390/jcm10102208_

Round 1

Reviewer 1 Report

In this paper entitled “Influence of Psychological Biomarkers on Therapeutic Adherence by Patients with Non-Alcoholic Fatty Liver Disease: A Moderated Mediation Model”, the authors evaluated the association between psychological biomarkers and therapeutic adherence among NAFLD patients, and concluded with the importance of social support and improvement self-efficacy which are related to better therapeutic adherence, which may become an important role in maintaining good control of this disease. I would like to add a few comments, which I believe would make this article better. Comment In this report, the authors pointed out that metabolic characteristic would impact the mental health of these patients, which is an important factor. Underlying conditions of NAFLD such as diabetes, obese, and SAS (sleep apnea syndrome) are said to have a strong correlation with depressive symptoms. However, the authors showed no clinical characteristics of the study population such as weight, BMI, comorbidity, and medications. Without this information, we could not evaluate the bias of certain background effecting the results of this study. I suggest that the authors should show the clinical background of the study population. Additionally, the authors showed the IPAQ-SF score for evaluation of the adherence to physical activity. However, they have not showed the range of score nor the index value of this score in the article, which leaves the readers not understanding how well the adherence is among the patients. The authors should show at least the index of this IPAQ-SF score, in order to make the readers understand what this score means.

Author Response

Response to Reviewer 1 Comments
Point 1:
In this paper entitled “Influence of Psychological Biomarkers on Therapeutic Adherence by Patients with Non-Alcoholic Fatty Liver Disease: A Moderated Mediation Model”, the authors evaluated the association between psychological biomarkers and therapeutic adherence among NAFLD patients, and concluded with the importance of social support and improvement self-efficacy which are related to better therapeutic adherence, which may become an important role in maintaining good control of this disease. I would like to add a few comments, which I believe would make this article better.
Response 1:
Thank you for your precise helpful suggestions.
Point 2:
In this report, the authors pointed out that metabolic characteristic would impact the mental health of these patients, which is an important factor. Underlying conditions of NAFLD such as diabetes, obese, and SAS (sleep apnea syndrome) are said to have a strong correlation with depressive symptoms. However, the authors showed no clinical characteristics of the study population such as weight, BMI, comorbidity, and medications. Without this information, we could not evaluate the bias of certain background effecting the results of this study. I suggest that the authors should show the clinical background of the study population.
Response 2:
We agree with your suggestions, and have added the analysis of the clinical characteristics of the participants: NASH, fibrosis, diabetes, BMI and obesity. As shown in Table 1, there were statistically significant differences in participants' therapeutic adherence by BMI and obesity. These variables were therefore included as covariates in the mediation and moderated mediation models. We feel this significantly improved the manuscript.
Point 3:
Additionally, the authors showed the IPAQ-SF score for evaluation of the adherence to physical activity. However, they have not showed the range of score nor the index value of this score in the article, which leaves the readers not understanding how well the adherence is among the patients. The authors should show at least the index of this IPAQ-SF score, in order to make the readers understand what this score means.
Response 3:
The International Physical Activity Questionnaire - Short Form (IPAQ-SF) measures the individual’s weekly physical activity, which is recorded in METs (metabolic equivalents for task). METs refer to the amount of energy a person expends while at rest. When performing some physical activity, the metabolic rate increases. The different Mets reflect the times basal metabolism increases in any activity. The IPAQ-SF records weekly physical activity in METs muliplied by the length in minutes of the activity and the number of days that week activity was done (METS X minutes X number of days a week). This was calculated using standardized METs according to the intensity of the activity:
- Walk: 3,3 METs.
- Moderate physical activity: 4 METs.
- Vigorous physical activity: 8 METs.
The score used in this study is the volume of total activity calculated by adding up the scores on each type of activity (walking + moderate physical activity + vigorous physical activity).
The total score (total physical activity) can be interpreted as a continuous variable or as a categorical variable. Due to the characteristics of the mediation analyses, we considered total physical activity as a continuous variable interpreted as: high scores indicate more physical activity (lines 131-132 in the manuscript). Each score is higher or lower depending on the weekly physical activity done by the individual. There are no rates or cutoff scores, and so they were not added in the study. We hope the Reviewer finds this explanation useful, and we are always willing to make any other addition or modification to the manuscript as required.

Reviewer 2 Report

The authors have investigated whether depressive symptoms mediated the association between physical quality of life and adherence to physical activity and the association between social support and adherence to diet in patients with NAFLD. Also, they evaluated whether self-efficacy exerted a moderating role in these associations.

Method

  1. When and for what reasons were the subjects received liver biopsy procedure?
  2. How about the results using Beck Depression Inventory II score?

Results

  1. Table 1: What was the result according to the severity of NAFLD (i.e presence of NASH or fibrosis)?
  2. Figure 2, 3,4) Please describe in the caption what the numbers in the figure mean. If it means indirect effect, is it more relevant in mediation as the number increases?

Minor

  1. It is better to combine Table 1 horizontally.
  2. In the method section 2.1 participants, the sentence starting with “Interventionary studies involving ---“ should be deleted

Author Response

Response to Reviewer 2 Comments
Point 1:
The authors have investigated whether depressive symptoms mediated the association between physical quality of life and adherence to physical activity and the association between social support and adherence to diet in patients with NAFLD. Also, they evaluated whether self-efficacy exerted a moderating role in these associations.
Response 1:
Thank you for your precise helpful suggestions and questions which we answer below.
Point 2:
Method. When and for what reasons were the subjects received liver biopsy procedure?
Response 2:
The study participants had been attending their respective hospitals for the few last years, and were referred from their corresponding primary care units, for probable liver damage. Patient liver pathology was confirmed by liver biopsy, and they were then included in the Spanish registry of NAFLD patients: HEPAmet. One of the inclusion criteria of this registry is NAFLD diagnosis by liver biopsy, so this is the main reason why prospective biopsies were performed to diagnose liver damage in patients.
Point 3:
Method. How about the results using Beck Depression Inventory II score?
Response 3:
In this study, we did not refer to the Beck Depression Inventory II (BDI-II) questionnaire. We used the Hospital Anxiety and Depression Scale (HADS) questionnaire to assess participants' depressive symptomatology, not the BDI-II.
Point 4:
Results. Table 1: What was the result according to the severity of NAFLD (i.e presence of NASH or fibrosis)?
Response 4:
Thank you for your suggestion. We have added NASH and fibrosis results in Table 1, as suggested.
Point 5:
Results. Figure 2, 3,4) Please describe in the caption what the numbers in the figure mean. If it means indirect effect, is it more relevant in mediation as the number increases?
Response 5:
Thank you for this correction. As suggested, we have described what the coefficients mean in the captions of Figures 2, 3, 4 and 6. Regarding the second question, these coefficients are unstandardized, so their significance levels and associated confidence intervals need to be checked to determine their importance.
Point 6:
Minor. It is better to combine Table 1 horizontally.
Response 6:
Thank you for this correction. As suggested, we have combined Table 1 horizontally. We feel this significantly improved the table.
Point 7:
Minor. In the method section 2.1 participants, the sentence starting with “Interventionary studies involving ---“ should be deleted
Response 7:
Thank you for your suggestion, we have deleted this sentence.

Round 2

Reviewer 1 Report

I would like to thank the authors for the rapid response and adjustment of the article.

With these additional information, I believe this article has become much more interesting, and also clinically useful.

Author Response

Response to Reviewer 1 Comments
Point 1:
I would like to thank the authors for the rapid response and adjustment of the article. With these additional information, I believe this article has become much more interesting, and also clinically useful.
Response 1:
Thank you for the encouraging comments on our manuscript.

Reviewer 2 Report

Fig legends 2.3.6: what is IMC?

Author Response

Response to Reviewer 2 Comments
Point 1:
Fig legends 2.3.6: what is IMC?
Response 1:
Thank you for this correction. We have put "BMI" instead of "IMC" in the legends of figures 2, 3, 4 and 6 to correctly refer to body mass index.
